# $O(T^{-1})$ Convergence of Optimistic-Follow-the-Regularized-Leader in Two-Player Zero-Sum Markov Games

**Yuepeng Yang**[*]                    **Cong Ma**[*]

## Abstract

We prove that optimistic-follow-the-regularized-leader (OFTRL), together with smooth value updates, finds an $O(T^{-1})$-approximate Nash equilibrium in $T$ iterations for two-player zero-sum Markov games with full information. This improves the $\tilde{O}(T^{-5/6})$ convergence rate recently shown in the paper by Zhang et al. (2022b). The refined analysis hinges on two essential ingredients. First, the sum of the regrets of the two players, though not necessarily non-negative as in normal-form games, is approximately non-negative in Markov games. This property allows us to bound the second-order path lengths of the learning dynamics. Second, we prove a tighter algebraic inequality regarding the weights deployed by OFTRL that shaves an extra $\log T$ factor. This crucial improvement enables the inductive analysis that leads to the final $O(T^{-1})$ rate.

## 1 Introduction

Multi-agent reinforcement learning (MARL) (Busoniu et al., 2008; Zhang et al., 2021) models sequential decision-making problems in which multiple agents/players interact with each other in a shared environment. MARL has recently achieved tremendous success in playing games (Vinyals et al., 2019; Berner et al., 2019; Brown & Sandholm, 2019), which, consequently, has spurred a growing body of work on MARL; see Yang & Wang (2020) for a recent overview.

A widely adopted mathematical model for MARL is the so-called Markov games (Shapley, 1953; Littman, 1994), which combines normal-form games (Nash, 1951) with Markov decision processes (Puterman, 2014). In a nutshell, a Markov game starts with a certain state, followed by actions taken by the players. The players then receive their respective payoffs, as in a normal-form game, and at the same time the system transits to a new state as in a Markov decision process. The whole process repeats. As in normal-form games, the goal for each player is to maximize her own cumulative payoffs. We defer the precise descriptions of Markov games to Section 2.

In the simpler normal-form games, no-regret learning (Cesa-Bianchi & Lugosi, 2006) has long been used as an effective method to achieve competence in the multi-agent environment. Take the two-player zero-sum normal-form game as an example. It is easy to show that standard no-regret algorithms such as follow-the-regularized-leader (FTRL) reach an $O(T^{-1/2})$-approximate Nash equilibrium (Nash, 1951) in $T$ iterations. Surprisingly, the seminal paper Daskalakis et al. (2011) demonstrates that a special no-regret algorithm, built upon Nesterov's excessive gap technique (Nesterov, 2005), achieves a faster and optimal $\tilde{O}(T^{-1})$ rate of convergence to the Nash equilibrium. This nice and fast convergence was later established for optimistic variants of mirror descent (Rakhlin & Sridharan, 2013) and FTRL (Syrgkanis et al., 2015). Since then,

---

[*]Department of Statistics, University of Chicago; Email: {yuepengyang, congm}@uchicago.edu

a flurry of research (Chen & Peng, 2020; Daskalakis et al., 2021; Anagnostides et al., 2022a;b; Farina et al., 2022) has been conducted around optimistic no-regret learning algorithms to obtain faster rate of convergence in normal-form games.

In contrast, research on the fast convergence of optimistic no-regret learning in Markov games has been scarce. In this paper, we focus on two-player zero-sum Markov games—arguably the simplest Markov game. Zhang et al. (2022b) recently initiated the study of the optimistic-follow-the-regularized-leader (OFTRL) algorithm in such a setting and proved that OFTRL converges to an $\tilde{O}(T^{-5/6})$-approximate Nash equilibrium after $T$ iterations. In light of the faster $O(T^{-1})$ convergence of optimistic algorithms in normal-form games, it is natural to ask

> *After $T$ iterations, can OFTRL find an $O(T^{-1})$-approximate Nash equilibrium in two-player zero-sum Markov games?*

In fact, this question has also been raised by Zhang et al. (2022b) in the Discussion section. More promisingly, they have verified the fast convergence (i.e., $O(T^{-1})$) of OFTRL in a simple two-stage Markov game; see Fig. 1 therein.

Our main contribution in this work is to answer this question *affirmatively*, through improving the $\tilde{O}(T^{-5/6})$ rate demonstrated in Zhang et al. (2022b) to the optimal $O(T^{-1})$ rate. The improved rate for OFTRL arises from two technical contributions. The first is the approximate non-negativity of the sum of the regrets of the two players in Markov games. In particular, the sum is lower bounded by the negative estimation error of the optimal $Q$-function; see Lemma 6 for the precise statement. This is in stark contrast to the two-player zero-sum normal-form game (Anagnostides et al., 2022c) and the multi-player general-sum normal-form game (Anagnostides et al., 2022b), in which by definition, the sum of the external/swap regrets are non-negative. This approximate non-negativity proves crucial for us to control the second-order path length of the learning dynamics induced by OFTRL. In a different context—time-varying zero-sum normal-form games, Zhang et al. (2022a) also utilizes a sort of approximate non-negativity of the sum of the regrets. However, the source of this gap from non-negativity is different: in Zhang et al. (2022a) it arises from the time-varying nature of the zero-sum game, while in our case with Markov games, it comes from the estimation error of the equilibrium pay-off matrix by the algorithm itself.

Secondly, central to the analysis in finite-horizon Markov decision processes (and also Markov games) is the induction across the horizon. In our case, in order to carry out the induction step, we prove a tighter algebraic inequality related to the weights deployed by OFTRL; see Lemma 4. In particular, we shave an extra $\log T$ factor. Surprisingly, this seemingly harmless $\log T$ factor is the key to enabling the above-mentioned induction analysis, and as a by-product, removes the extra $\log$ factor in the performance guarantee of OFTRL.

Note that as an imperfect remedy, Zhang et al. (2022b) proposed a modified OFTRL algorithm that achieves $\tilde{O}(T^{-1})$ convergence to Nash equilibrium. However, compared to the vanilla OFTRL algorithm considered herein, the modified version tracks two $Q$-functions, adopts a different $Q$-function update procedure that can be more costly in certain scenarios, and more importantly diverges from the general policy optimization framework proposed in Zhang et al. (2022b). Our work bridges these gaps by establishing the fast convergence for the vanilla OFTRL.

Another line of algorithms used for solving Nash equilibrium is based on dynamic programming (Perolat et al., 2015; Zhang et al., 2022b; Cen et al., 2021). Unlike the single-loop structure of OFTRL, the dynamic programming approach requires a nested loop, with the outer-loop iterating over the horizons and the inner-loops solving a sub-game through iterations. This requires more tuning parameters, one set for each subproblem/layer. Such kind of extra tuning was documented in Cen et al. (2021). The nested nature of dynamic programming also demands one to predetermine a precision $\epsilon$ and estimate the sub-game at each horizon to precision $\epsilon/H$. This is less convenient in practice compared to a single-loop algorithm like the

OFTRL we study, where such predetermined precision is not necessary. Another recent paper Cen et al. (2022) also discusses the advantages of single-loop algorithms over those with nested loops.

## 1.1 RELATED WORK

**Optimistic no-regret learning in games.** Our work is mostly related to the line of work on proving fast convergence of optimistic no-regret algorithms in various forms of games. Daskalakis et al. (2011) provide the first fast algorithm that reaches a Nash equilibrium at an $\tilde{O}(T^{-1})$ rate in two-player zero-sum normal-form games. Later, with the same setup, Rakhlin & Sridharan (2013) prove a similar fast convergence for optimistic mirror descent (OMD). Syrgkanis et al. (2015) extend the results to multi-player general-sum normal-form games. In addition, Syrgkanis et al. show that when all the players adopt optimistic algorithms, their *individual* regret is at most $O(T^{-3/4})$. This is further improved to $O(T^{-5/6})$ in the special two-player zero-sum case (Chen & Peng, 2020). More recently, via a detailed analysis of higher-order smoothness, Daskalakis et al. (2021); Anagnostides et al. (2022a) manage to improve the individual regret guarantee of optimistic hedge to $\tilde{O}(T^{-1})$ in multi-player general-sum normal-form games, matching the result in the two-player case. A similar result is shown by Anagnostides et al. (2022b) with a different analysis using self-concordant barriers as the regularizer.

Several attempts have been made to extend the results on optimistic no-regret learning in normal-form games to Markov games. Wei et al. (2021) design a decentralized algorithm based on optimistic gradient descent / ascent that converges to a Nash equilibrium at an $\tilde{O}(T^{-1/2})$ rate. Closest to us is the work by Zhang et al. (2022b) which shows an $\tilde{O}(T^{-5/6})$ convergence of OFTRL to the Nash equilibrium in two-player zero-sum Markov games and an $\tilde{O}(T^{-3/4})$ convergence to a coarse correlated equilibrium in multi-player general-sum Markov games. Most recently, Erez et al. (2022) prove an $O(T^{-1/4})$ individual regret for OMD in multi-player general-sum Markov games.

**Two-player zero-sum Markov games.** Our work also fits into the study of two-player zero-sum Markov games (Shapley, 1953; Littman, 1994). Various algorithms (Hu & Wellman, 2003; Littman, 1994; Zhao et al., 2021; Cen et al., 2021) have been proposed in the full information setting, where one assumes the players have access to the *exact* state-action value functions. In particular, Zhao et al. (2021); Cen et al. (2021) use optimistic approaches for normal-form games as subroutines to extend the $\tilde{O}(T^{-1})$ convergence rates to two-player zero-sum Markov games. In particular, they provide last iterate convergence guarantees as well. However, in doing so, their algorithms require one to approximately solve a normal-form game in each iteration.

In the bandit setting, Bai & Jin (2020); Xie et al. (2020); Bai et al. (2020); Liu et al. (2021); Zhang et al. (2020) study the sample complexity of two-player zero-sum Markov games. In addition, Sidford et al. (2020); Jia et al. (2019); Zhang et al. (2020); Li et al. (2022) investigate the sample complexity under a generative model where one can query the Markov game at arbitrary states and actions. Last but not least, recently two-player zero-sum Markov games have been studied in the offline setting (Cui & Du, 2022; Yan et al., 2022), where the learner is given a set of historical data, and cannot interact with Markov games further.

## 2 PRELIMINARIES

This section provides the necessary background on Markov games and optimistic-follow-the-regularized-leader (OFTRL).

**Two-player zero-sum Markov games.** Denote by $\mathcal{MG}(H, \mathcal{S}, \mathcal{A}, \mathcal{B}, \mathbb{P}, r)$ a finite-horizon time-inhomogeneous two-player zero-sum Markov game, with $H$ the horizon, $\mathcal{S}$ the state space, $\mathcal{A}$ (resp. $\mathcal{B}$) the action space for the max-player (resp. min-player), $\mathbb{P} = \{\mathbb{P}_h\}_{h\in[H]}$ the transition probabilities, and $r = \{r_h\}_{h\in[H]}$ the reward function. We assume state space $\mathcal{S}$ and action spaces $\mathcal{A}, \mathcal{B}$ to be finite and have size $S, A, B$, respectively, and $r_h$ takes value in $[0, 1]$. Without loss of generality, we assume that the game starts at a fixed state $s_1 \in \mathcal{S}$. Then at each step $h$, both players observe the current state $s_h \in \mathcal{S}$. The max-player picks an action $a_h \in \mathcal{A}$ and the min-player picks an action $b_h \in \mathcal{B}$ simultaneously. Then the max-player (resp. min-player) receives the reward $r_h(s_h, a_h, b_h)$ (resp. $-r_h(s_h, a_h, b_h)$), and the game transits to step $h+1$ with the next state $s_{h+1}$ sampled from $\mathbb{P}_h(\cdot \mid s_h, a_h, b_h)$. The game ends after $H$ steps. The goal for the max-player is to maximize her total reward while the min-player seeks to minimize the total reward obtained by the max-player.

**Markov policies and value functions.** Let $\mu = \{\mu_h\}_{h\in[H]}$ be the Markov policy for the max-player, where $\mu_h(\cdot \mid s) \in \Delta_{\mathcal{A}}$ is the distribution of actions the max-player picks when seeing state $s$ at step $h$. Here, $\Delta_{\mathcal{X}}$ denotes the set of all probability distributions on the space $\mathcal{X}$. Similarly, the min-player is equipped with a Markov policy $\nu = \{\nu_h\}_{h\in[H]}$. We define the value function of the policy pair $(\mu, \nu)$ at step $h$ to be

$$V_h^{\mu,\nu}(s) := \mathbb{E}_{\mu,\nu}\left[\sum_{i=h}^{H} r(s_i, a_i, b_i) \mid s_h = s\right],$$

where the expectation is taken w.r.t. the policies $\{\mu_i, \nu_i\}_{i\geq h}$ and the state transitions $\{\mathbb{P}_i\}_{i\geq h}$. Similarly, one can define the $Q$-function as

$$Q_h^{\mu,\nu}(s, a, b) := \mathbb{E}_{\mu,\nu}\left[\sum_{i=h}^{H} r(s_i, a_i, b_i) \mid s_h = s, a_h = a, b_h = b\right].$$

In words, both functions represent the expected future rewards received by the max-player given the current state or state-action pair.

**Best responses and Nash equilibria.** Fix a Markov policy $\nu$ for the min-player. There exists a Markov policy $\mu^{\dagger}(\nu)$ (a.k.a. best response) such that for any $s \in \mathcal{S}$ and $h \in [H]$,

$$V_h^{\mu^{\dagger}(\nu),\nu}(s) = \sup_{\mu^{\dagger}} V_h^{\mu^{\dagger},\nu}(s),$$

where the supremum is taken over all Markov policies. To simplify the notation, we denote $V_h^{\dagger,\nu}(s) := V_h^{\mu^{\dagger}(\nu),\nu}(s)$. Similarly, we can define $V_h^{\mu,\dagger}(s)$. It is known that a pair $(\mu^{\star}, \nu^{\star})$ of Markov policies exists and $\mu^{\star}, \nu^{\star}$ are best responses to the other, i.e., $V_h^{\mu^{\star},\nu^{\star}}(s) = V_h^{\dagger,\nu^{\star}}(s) = V_h^{\mu^{\star},\dagger}(s)$ for all $s \in \mathcal{S}$ and $h \in [H]$. Such a pair $(\mu^{\star}, \nu^{\star})$ is called a Nash equilibrium (NE). We may denote the value function and $Q$-function under any Nash equilibrium $(\mu^{\star}, \nu^{\star})$ as

$$V_h^{\star} := V_h^{\mu^{\star},\nu^{\star}}, \qquad Q_h^{\star} := Q_h^{\mu^{\star},\nu^{\star}},$$

which are known to be unique even if there are multiple Nash equilibria (Shapley, 1953). The goal of learning in two-player zero-sum Markov games is to find an $\varepsilon$-approximation to the NE defined as follows.

**Definition 1** ($\varepsilon$-approximate Nash equilibrium). *Fix any approximation accuracy $\varepsilon > 0$. A pair $(\mu, \nu)$ of Markov policies is an $\varepsilon$-approximate Nash equilibrium if*

$$\text{NE-gap}(\mu, \nu) := V_1^{\dagger,\nu}(s_1) - V_1^{\mu,\dagger}(s_1) \leq \varepsilon. \tag{1}$$

---

**Algorithm 1** Optimistic-follow-the-regularized-leader for solving two-player zero-sum Markov games

---

**Input:** Stepsize $\eta$, reward function $r$, probability transition function $\mathbb{P}$.
**Initialization:** $Q_h^0 \equiv 0$ for all $h \in [H]$.
**For iteration** 1 **to** $T$, **do**

- **Policy Update:** for all state $s \in \mathcal{S}$, horizon $h \in [H]$,

$$\mu_h^t(a \mid s) \propto \exp\left(\frac{\eta}{w_t}\left[\sum_{i=1}^{t-1} w_i \left[Q_h^i \nu_h^i\right](s,a) + w_t \left[Q_h^{t-1} \nu_h^{t-1}\right](s,a)\right]\right), \quad (2a)$$

$$\nu_h^t(b \mid s) \propto \exp\left(-\frac{\eta}{w_t}\left[\sum_{i=1}^{t-1} w_i \left[(Q_h^i)^\top \mu_h^i\right](s,b) + w_t \left[(Q_h^{t-1})^\top \mu_h^{t-1}\right](s,b)\right]\right). \quad (2b)$$

- **Value Update:** for all $s \in \mathcal{S}, a \in \mathcal{A}, b \in \mathcal{B}$, from $h = H$ to 1,

$$Q_h^t(s,a,b) = (1-\alpha_t)Q_h^{t-1}(s,a,b) + \alpha_t\left(r_h + \mathbb{P}_h\left[(\mu_{h+1}^t)^\top Q_{h+1}^t \nu_{h+1}^t\right]\right)(s,a,b), \quad (3)$$

**Output average policy:** for all $s \in \mathcal{S}, h \in [H]$

$$\hat{\mu}_h(\cdot \mid s) := \sum_{t=1}^T \alpha_T^t \mu_h^t(\cdot \mid s), \quad \hat{\nu}_h(\cdot \mid s) := \sum_{t=1}^T \alpha_T^t \nu_h^t(\cdot \mid s). \quad (4)$$

---

**An interlude: additional notations.** Before explaining OFTRL, we introduce some additional notations to simplify things hereafter. Fix any $h \in [H]$, $s \in \mathcal{S}$. For any function $Q : \mathcal{S} \times \mathcal{A} \times \mathcal{B} \to \mathbb{R}$, we may consider $Q(s, \cdot, \cdot)$ to be an $A \times B$ matrix and $\mu_h(\cdot \mid s), \nu_h(\cdot \mid s)$ to be vectors of length $A$ and $B$, respectively. Then for any policy $(\mu_h, \nu_h)$ at horizon $h$ we may define

$$\left[\mu_h^\top Q \nu_h\right](s) := \mathbb{E}_{a \sim \mu_h(\cdot \mid s), b \sim \nu_h(\cdot \mid s)}[Q(s,a,b)],$$
$$\left[\mu_h^\top Q\right](s, \cdot) := \mathbb{E}_{a \sim \mu_h(\cdot \mid s)}[Q(s,a,\cdot)],$$
$$\left[Q \nu_h\right](s, \cdot) := \mathbb{E}_{b \sim \nu_h(\cdot \mid s)}[Q(s,\cdot,b)].$$

The term $\left[\mu_h^\top Q \nu_h\right](s)$ can also be written in the inner product form $\langle \mu_h, Q\nu_h \rangle(s)$ or $\langle \nu_h, Q^\top \mu_h \rangle(s)$. It is easy to check that for fixed $s$ and $h$, the left hand sides of these definitions are standard matrix operations. In addition, for any $V : \mathcal{S} \mapsto \mathbb{R}$, we define the shorthand

$$\left[\mathbb{P}_h V\right](s,a,b) := \mathbb{E}_{s' \sim \mathbb{P}_h(\cdot \mid s,a,b)}[V(s')],$$

which allows us to rewrite Bellman updates of $V$ and $Q$ as

$$V_h^{\mu,\nu}(s) = \left[\mu_h^\top Q_h^{\mu,\nu} \nu_h\right](s),$$
$$Q_h^{\mu,\nu}(s,a,b) = r_h(s,a,b) + \left[\mathbb{P}_h V_{h+1}^{\mu,\nu}\right](s,a,b).$$

**Optimistic-follow-the-regularized-leader.** Now we are ready to introduce the optimistic-follow-the-regularized-leader (OFTRL) algorithm for solving two-player zero-sum Markov games, which has appeared in the paper by Zhang et al. (2022b). See Algorithm 1 for the full specification.

In a nutshell, the algorithm has three main components. The first is the policy update (2) using *weighted OFTRL* for both the max and min players. As one can see, compared to the standard follow-the-regularized-leader algorithm, the *weighted OFTRL* adds a loss predictor $[Q^{t-1}\nu^{t-1}](s,a)$ and deploys a weighted update

according to the weights $\{w_i\}_{1 \le i \le t}$, which we shall define momentarily. The second component is the backward value update (3) using weighted average of the previous estimates and the Bellman updates. The last essential part is outputting a weighted policy (4) over all the historical policies. As one can realize, weights play a big role in specifying the OFTRL algorithm. In particular, we set

$$\alpha_t := \frac{H+1}{H+t}, \quad \alpha_t^t := \alpha_t, \quad \alpha_t^i := \alpha_i \prod_{j=i+1}^{t} (1 - \alpha_j), \quad w_i := \frac{\alpha_t^i}{\alpha_t^1} = \frac{\alpha_i}{\alpha_1 \prod_{j=2}^{i}(1 - \alpha_j)}, \quad (5)$$

which are the same choices as in the paper by (Zhang et al., 2022b).

## 3 MAIN RESULT AND OVERVIEW OF THE PROOF

With the preliminaries in place, we are in a position to state our main result for OFTRL in two-player zero-sum Markov games.

**Theorem 1.** *Consider Algorithm 1 with $\eta = C_\eta H^{-2}$ for some constant $C_\eta \le 1/8$. The output policy pair $(\hat{\mu}, \hat{\nu})$ satisfies*

$$\text{NE-gap}(\hat{\mu}, \hat{\nu}) \le \frac{320 C_\eta^{-1} H^5 \cdot \log(AB)}{T}.$$

Several remarks on Theorem 1 are in order. First, Theorem 1 demonstrates that OFTRL can find an $O(T^{-1})$-approximate Nash equilibrium in $T$ iterations. This improves the $\tilde{O}(T^{-5/6})$ rate proved in the prior work (Zhang et al., 2022b), and also matches the empirical evidence provided therein. While the paper by Zhang et al. (2022b) also provides a modified OFTRL algorithm that achieves an $\tilde{O}(T^{-1})$ rate by maintaining two separate value estimators (one for the max-player and the other for the min-player), the OFTRL algorithm studied herein is more natural and also computationally simpler. Second, this rate is nearly unimprovable even in the simpler two-player zero-sum normal-form games (Daskalakis et al., 2011). It is also worth pointing out that algorithms with $\tilde{O}(T^{-1})$ rate have been proposed in the literature (Cen et al., 2021; Zhao et al., 2021). However, compared to those algorithms, OFTRL does not require one to approximately solve a normal-form game in each iteration. Lastly, Theorem 1 allows any $C_\eta \in (0, 1/8]$ while $C_\eta = 1/8$ is optimal for the bound on NE-gap.

Before embarking on the formal proof, we would like to immediately provide an overview of our proof techniques.

**Step 1: controlling NE-gap using the sum of regrets and estimation error.** In the simpler normal-form game (i.e., without any state transition dynamics as in Markov games), it is well known that NE-gap is controlled by the sum of the regrets of the two players. This would also be the case for Markov games if in the policy update (2) by OFTRL, we use the true $Q$-function $Q_h^{t}$ instead of the estimate $Q_h^t$. As a result, intuitively, the NE-gap in Markov games should be controlled by both the sum of the regrets of the two players and also the estimation error $\|Q_h^t - Q_h^\star\|_\infty$; see Lemma 1.

**Step 2: bounding the sum of regrets.** Given the extensive literature on regret guarantees for optimistic algorithms (Anagnostides et al., 2022c;b; Zhang et al., 2022b), it is relatively easy to control the sum of the regrets to obtain the desired $O(T^{-1})$ rate; see Lemma 2. The key is to exploit the stability in the loss vectors.

**Step 3: bounding estimation error.** It then boils down to controlling the estimation error $\|Q_h^t - Q_h^\star\|_\infty$, in which our main technical contributions lie. Due to the nature of the Bellman update (3), it is not hard to obtain a recursive relation for the estimation error; see the recursion (17). However, the undesirable part is that the estimation error depends on the *maximal* regret between the two players, instead of the *sum* of the

regrets. This calls for technical innovation. Inspired by the work of Anagnostides et al. (2022c;b) in normal-form games, we make an important observation that the sum of the regrets is approximately non-negative. In particular, the sum is lower bounded by the negative estimation error $\|Q_h^t - Q_h^\star\|_\infty$; see Lemma 6. This lower bound together with the upper bound in Step 2 allows us to control the maximal regret via the estimation error (19), which further yields a recursive relation (20) involving estimation errors only. Solving the recursion leads to the desired result.

## 4 PROOF OF THEOREM 1

In this section, we present the proof of our main result, i.e., Theorem 1. We first define a few useful notations. For each step $h \in [H]$, each state $s \in \mathcal{S}$, and each iteration $t \in [T]$, we define the state-wise weighted individual regret as

$$\text{reg}_{h,1}^t(s) := \max_{\mu^\dagger \in \Delta_\mathcal{A}} \sum_{i=1}^t \alpha_t^i \left\langle \mu^\dagger - \mu_h^i, Q_h^i \nu_h^i \right\rangle (s), \tag{6a}$$

$$\text{reg}_{h,2}^t(s) := \max_{\nu^\dagger \in \Delta_\mathcal{B}} \sum_{i=1}^t \alpha_t^i \left\langle \nu_h^i - \nu^\dagger, (Q_h^i)^\top \mu_h^i \right\rangle (s). \tag{6b}$$

We also define the maximal regret as

$$\text{reg}_h^t := \max_{s \in \mathcal{S}} \max_{i=1,2} \left\{ \text{reg}_{h,i}^t(s) \right\},$$

that maximizes over the players and the states. In addition, for each step $h \in [H]$, and each iteration $t \in [T]$, we define the estimation error of the $Q$-function as

$$\delta_h^t := \|Q_h^t - Q_h^\star\|_\infty.$$

With these notations in place, we first connect the NE-gap with the sum of regrets $\text{reg}_{h,1}^T(s) + \text{reg}_{h,2}^T(s)$ as well as the estimation error $\delta_h^t$.

**Lemma 1.** *One has*

$$\text{NE-gap}(\hat\mu, \hat\nu) \leq 2 \sum_{h=1}^H \left\{ \max_s \left\{ \text{reg}_{h,1}^T(s) + \text{reg}_{h,2}^T(s) \right\} + 2 \sum_{t=1}^T \alpha_T^t \delta_h^t \right\}.$$

See Section B.1 for the proof of this lemma.

It then boils down to controlling $\max_s \left\{ \text{reg}_{h,1}^T(s) + \text{reg}_{h,2}^T(s) \right\}$ and $\sum_{t=1}^T \alpha_T^t \delta_h^t$. The following two lemmas provide such control.

**Lemma 2.** *For every $h \in [H]$, every $s \in \mathcal{S}$, and every iteration $t \in [T]$, one has*

$$\mathrm{reg}_{h,1}^t(s) \leq \frac{2H \cdot (\log A)}{\eta t} + \frac{16\eta H^3}{t} + 2\eta H^2 \sum_{i=2}^{t} \alpha_t^i \|\nu_h^i(\cdot \mid s) - \nu_h^{i-1}(\cdot \mid s)\|_1^2 \tag{7a}$$

$$- \frac{1}{8\eta} \sum_{i=2}^{t} \alpha_t^{i-1} \|\mu_h^i(\cdot \mid s) - \mu_h^{i-1}(\cdot \mid s)\|_1^2;$$

$$\mathrm{reg}_{h,2}^t(s) \leq \frac{2H \cdot (\log B)}{\eta t} + \frac{16\eta H^3}{t} + 2\eta H^2 \sum_{i=2}^{t} \alpha_t^i \|\mu_h^i(\cdot \mid s) - \mu_h^{i-1}(\cdot \mid s)\|_1^2 \tag{7b}$$

$$- \frac{1}{8\eta} \sum_{i=2}^{t} \alpha_t^{i-1} \|\nu_h^i(\cdot \mid s) - \nu_h^{i-1}(\cdot \mid s)\|_1^2.$$

*As a result, when $\eta = C_\eta H^{-2}$ for some constant $C_\eta \leq 1/8$, one has*

$$\max_s \left\{ \mathrm{reg}_{h,1}^t(s) + \mathrm{reg}_{h,2}^t(s) \right\} \leq \frac{3C_\eta^{-1} H^3 \cdot \log(AB)}{t} - 4\eta H^3 \sum_{i=2}^{t} \alpha_t^i \Big( \|\mu_h^i(\cdot \mid s) - \mu_h^{i-1}(\cdot \mid s)\|_1^2 \tag{8}$$

$$+ \|\nu_h^i(\cdot \mid s) - \nu_h^{i-1}(\cdot \mid s)\|_1^2 \Big).$$

See Section B.2 for the proof of this lemma.

**Lemma 3.** *Choosing $\eta = C_\eta H^{-2}$ for some constant $C_\eta \leq 1/8$, for all $h \in [H]$ and $t \in [T]$, we have that*

$$\delta_h^t \leq \frac{5e^2 C_\eta^{-1} H^4 \cdot \log(AB)}{t}.$$

See Section B.3 for the proof of this lemma.

Combine Lemmas 2-3 with Lemma 1 to arrive at the desired conclusion that when $\eta = C_\eta H^{-2}$ for some constant $C_\eta \leq 1/8$,

$$\mathrm{NE\text{-}gap}(\hat{\mu}, \hat{\nu}) \leq 2 \sum_{h=1}^{H} \left\{ \max_s \left\{ \mathrm{reg}_{h,1}^T(s) + \mathrm{reg}_{h,2}^T(s) \right\} + 2 \sum_{t=1}^{T} \alpha_T^t \delta_h^t \right\}$$

$$\leq 2 \sum_{h=1}^{H} \left\{ \frac{3C_\eta^{-1} H^3 \cdot \log(AB)}{T} + 2 \sum_{t=1}^{T} \alpha_T^t \frac{5e^2 C_\eta^{-1} H^4 \cdot \log(AB)}{t} \right\}$$

$$\leq 2H \cdot \left\{ \frac{3C_\eta^{-1} H^3 \cdot \log(AB)}{T} + \frac{20e^2 C_\eta^{-1} H^4 \cdot \log(AB)}{T} \right\}$$

$$\leq \frac{320 C_\eta^{-1} H^5 \cdot \log(AB)}{T},$$

where the penultimate inequality uses the following important lemma we have alluded to before.

**Lemma 4.** *For all $t \geq 1$, one has*

$$\sum_{i=1}^{t} \alpha_t^i \cdot \frac{1}{i} \leq \left( 1 + \frac{1}{H} \right) \frac{1}{t}. \tag{9}$$

On the surface, this lemma shaves an extra $\log t$ factor from a simple average of the sequence $\{1/i\}_{i \le t}$ (cf. Lemma A.3 in the paper by Zhang et al. (2022b)). But more importantly, it shines in the ensuing proof of Lemma 3 by enabling the induction step. See Section B.4 for the proof of Lemma 4, and see the end of Section B.3 for the comment on the benefit of this improved result.

## 5 DISCUSSION

In this paper, we prove that the optimistic-follow-the-regularized-leader algorithm, together with smooth value updates, converges to an $O(T^{-1})$-approximate Nash equilibrium in two-player zero-sum Markov games. This improves the $\tilde{O}(T^{-5/6})$ rate proved in the paper Zhang et al. (2022b). Quite a few interesting directions are open. Below we single out a few of them. First, although our rate is unimprovable in the dependence on $T$, it is likely sub-optimal in its dependence on the horizon $H$. Improving such dependence and proving any sort of lower bound on it are both interesting and important for finite-horizon Markov games. Second, we focus on the simple two-player zero-sum games. It is an important open question to see whether one can generalize the proof technique herein to the multi-player general-sum Markov games and to other solution concepts in games (e.g., coarse correlated equilibria, and correlated equilibria).

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

## A    PROPERTIES OF $\alpha_t^i$

This section collects a few useful properties of the sequences $\{\alpha_t\}_{t \geq 1}$ and $\{\alpha_t^i\}_{t \geq 1, 1 \leq i \leq t}$. Some of these results have appeared in prior work (Jin et al., 2018; Zhang et al., 2022b). For completeness, we include all the proofs here.

To help reading, we repeat the definitions below: for each $t \geq 1$, and $1 \leq i \leq t$, we define

$$\alpha_t = \alpha_t^t = \frac{H+1}{H+t}, \qquad \text{and} \tag{10a}$$

$$\alpha_t^i = \alpha_i \prod_{j=i+1}^{t} (1 - \alpha_j). \tag{10b}$$

**Lemma 5.** *Fix any $t \geq 1$. The following properties are true:*

1. *The sequence $\{\alpha_t^i\}_{1 \leq i \leq t}$ sums to 1, i.e., $\sum_{i=1}^{t} \alpha_t^i = 1$.*

2. *For all $1 \leq i \leq t$, one has $\alpha_t^i \leq i/t$.*

3. *For the relative weight defined by $w_i = \alpha_t^i/\alpha_t^1$ (note that this is the same for every $t \geq i$), we have*

$$\frac{w_i}{w_{i-1}} = \frac{\alpha_t^i}{\alpha_t^{i-1}} = \frac{H+i-1}{i-1} \leq H.$$

4. *The sequence $\{\alpha_t^i\}_{1 \leq i \leq t}$ is increasing in $i$.*

5. *On the sum of squares of the weights, we have*

$$\sum_{i=1}^{t} (\alpha_t^i)^2 \leq \sum_{i=1}^{t} \alpha_i^2 \leq H + 2.$$

6. *For any non-increasing sequence $\{b_i\}_{1 \leq i \leq t}$, one has*

$$\sum_{i=1}^{t} \alpha_t^i b_i \leq \frac{1}{t} \sum_{i=1}^{t} b_i.$$

*Proof.* Property 1 follows directly from the definitions of $\left\{\alpha_t^i\right\}_{1 \leq i \leq t}$.

Now we move on to Property 2. It trivially holds for $i = t$. Therefore we focus on the case when $1 \leq i \leq t-1$. By definition, we have

$$\alpha_t^i = \alpha_i \prod_{j=i+1}^{t} (1 - \alpha_j) \leq \prod_{j=i+1}^{t} (1 - \alpha_j) = \prod_{j=i+1}^{t} \frac{j-1}{H+j}. \tag{11}$$

where the inequality holds since $\alpha_i \leq 1$ for all $1 \leq i \leq t$, and the last relation is the definition of $\alpha_j$. Expanding the right hand side of (11), we have

$$\alpha_t^i \leq \frac{i}{H+i+1} \times \frac{i+1}{H+i+2} \times \cdots \times \frac{t-1}{H+t} \leq \frac{i}{H+t},$$

where we only keep the first numerator and the last denominator. Property 2 then follows.

Property 3 is trivial. Hence we omit the proof. In addition, Property 3 implies Property 4 since $\frac{\alpha_t^i}{\alpha_t^{i-1}} = \frac{H+i-1}{i-1} \geq 1$.

For Property 5, the first inequality holds since $0 \le \alpha_i \le 1$ for all $1 \le i \le t$. For the second inequality, one has

$$\sum_{i=1}^{t} \alpha_i^2 = 1 + \sum_{i=2}^{t} \left( \frac{H+1}{H+i} \right)^2 \le 1 + (H+1)^2 \sum_{i=2}^{t} \left( \frac{1}{(H+i-1)(H+i)} \right).$$

Expanding this as a telescoping sum, we see that

$$\sum_{i=1}^{t} \alpha_i^2 \le 1 + (H+1)^2 \sum_{i=2}^{t} \left( \frac{1}{H+i-1} - \frac{1}{H+i} \right)$$
$$\le 1 + (H+1)^2 \frac{1}{H+1}$$
$$= H + 2.$$

Lastly, for Property 6, we have

$$\sum_{i=1}^{t} \alpha_t^i b_i - \frac{1}{t} \sum_{i=1}^{t} b_i = \sum_{i=1}^{t} (\alpha_t^i - \frac{1}{t}) b_i.$$

Let $i_0 := \sup_i \left\{ \alpha_t^i \le 1/t \right\}$. Since $\{\alpha_t^i\}$ is increasing in $i$ (cf. Property 4) and $\sum_{i=1}^{t} \alpha_t^i = 1$ (cf. Property 1), we know that $i_0$ is well defined, i.e., $1 \le i_0 \le t$. Since $\{\alpha_t^i\}_{i \le t}$ (resp. $\{b_i\}_{i \le t}$) is increasing (resp. non-increasing), we have $\alpha_t^i \le 1/t$ and $b_i \ge b_{i_0}$ for all $i \le i_0$. As a result, we obtain $(\alpha_t^i - 1/t) b_i \le (\alpha_t^i - 1/t) b_{i_0}$ for all $i \le i_0$. Similarly, one has $\alpha_t^i > 1/t$ and $b_i \le b_{i_0}$ for all $i > i_0$, which implies $(\alpha_t^i - 1/t) b_i \le (\alpha_t^i - 1/t) b_{i_0}$ for all $i > i_0$. Take these two relations together to see that

$$\sum_{i=1}^{t} (\alpha_t^i - 1/t) b_i \le \sum_{i=1}^{t} (\alpha_t^i - 1/t) b_{i_0} = 0,$$

where the last equality uses the fact from Property 1, namely $\sum_{i=1}^{t} \alpha_t^i = 1$.

## B    PROOF OF SUPPORTING LEMMAS IN SECTION 4

### B.1    PROOF OF LEMMA 1

Invoke Lemma C.1 in the paper by Zhang et al. (2022b) to obtain

$$\text{NE-gap}(\hat{\mu}, \hat{\nu}) = V_1^{\dagger, \hat{\nu}}(s_1) - V_1^{\star}(s_1) + V_1^{\star}(s_1) - V_1^{\hat{\mu}, \dagger}(s_1)$$
$$\le 2 \sum_{h=1}^{H} \max_s \left\{ \max_{\mu^\dagger, \nu^\dagger} \left[ \langle \mu^\dagger, Q_h^\star \hat{\nu}_h \rangle - \langle \nu^\dagger, Q_h^{\star\top} \hat{\mu}_h \rangle \right](s) \right\}.$$

By the definition of the output policy $(\hat{\mu}, \hat{\nu})$, one has

$$\max_{\mu^\dagger, \nu^\dagger} \left[ \langle \mu^\dagger, Q_h^\star \hat{\nu}_h \rangle - \langle \nu^\dagger, Q_h^{\star\top} \hat{\mu}_h \rangle \right](s) = \max_{\mu^\dagger, \nu^\dagger} \sum_{t=1}^{T} \alpha_T^t \left[ \langle \mu^\dagger, Q_h^\star \nu_h^t \rangle - \langle \nu^\dagger, Q_h^{\star\top} \mu_h^t \rangle \right](s).$$

Replacing the true value function $Q_h^\star$ with the value estimate $Q_h^t$ yields

$$\max_{\mu^\dagger, \nu^\dagger} \left[ \langle \mu, Q_h^\star \hat{\nu}_h \rangle - \langle \nu^\dagger, (Q_h^\star)^\top \hat{\mu}_h \rangle \right](s) \leq \max_{\mu^\dagger, \nu^\dagger} \sum_{t=1}^{T} \alpha_T^t \left[ \langle \mu^\dagger, Q_h^t \nu_h^t \rangle - \langle \nu^\dagger, (Q_h^t)^\top \mu_h^t \rangle \right](s) + 2 \sum_{t=1}^{T} \alpha_T^t \delta_h^t,$$

where we recall $\delta_h^t = \|Q_h^t - Q_h^\star\|_\infty$. The proof is finished by taking the above three relations together with the observation that

$$\mathrm{reg}_{h,1}^T(s) + \mathrm{reg}_{h,2}^T(s) = \max_{\mu^\dagger, \nu^\dagger} \sum_{t=1}^{T} \alpha_T^t \left[ \langle \mu^\dagger, Q_h^t \nu_h^t \rangle - \langle \nu^\dagger, (Q_h^t)^\top \mu_h^t \rangle \right](s).$$

## B.2 Proof of Lemma 2

We prove the regret bound for the max-player (i.e., bound (7a)). The bound (7b) for the min-player can be obtained via symmetry.

First, we make the observation that, the policy update in Algorithm 1 for the max-player is exactly the OFTRL algorithm (i.e., Algorithm 4 in the paper by Zhang et al. (2022b)) with the loss vector $g_t = w_t [Q_h^t \nu_h^t](s, \cdot)$, the recency bias $M_t = w_t [Q_h^{t-1} \nu_h^{t-1}](s, \cdot)$, and a learning rate $\eta_t = \eta/w_t$. Therefore, we can apply Lemma B.3 from Zhang et al. (2022b) to obtain

$$\mathrm{reg}_{h,1}^t(s) = \max_{\mu^\dagger} \sum_{i=1}^{t} \alpha_t^i \left\langle (\mu^\dagger - \mu_h^i), Q_h^i \nu_h^i \right\rangle(s)$$

$$= \alpha_t^1 \max_{\mu^\dagger} \sum_{i=1}^{t} w_i \left\langle (\mu^\dagger - \mu_h^i), Q_h^i \nu_h^i \right\rangle(s)$$

$$\leq \frac{\alpha_t \cdot (\log A)}{\eta} + \alpha_t^1 \underbrace{\sum_{i=1}^{t} \frac{\eta}{w_i} \left\| \left[ w_i Q_h^i \nu_h^i - w_i Q_h^{i-1} \nu_h^{i-1} \right](s, \cdot) \right\|_\infty^2}_{=: \mathrm{Err}_1} \tag{12}$$

$$- \alpha_t^1 \underbrace{\sum_{i=2}^{t} \frac{w_{i-1}}{8\eta} \|\mu_h^i(\cdot \mid s) - \mu_h^{i-1}(\cdot \mid s)\|_1^2}_{=: \mathrm{Err}_2}, \tag{13}$$

where we have used the fact that $w_i = \alpha_t^i/\alpha_t^1$. We now move on to bound the term $\mathrm{Err}_1$. Use $(a+b)^2 \leq 2a^2 + 2b^2$ to see that

$$\left\| \left[ Q_h^i \nu_h^i - Q_h^{i-1} \nu_h^{i-1} \right](s, \cdot) \right\|_\infty^2 \leq 2 \left\| \left[ Q_h^i \nu_h^i - Q_h^{i-1} \nu_h^i \right](s, \cdot) \right\|_\infty^2 + 2 \left\| \left[ Q_h^{i-1} \nu_h^i - Q_h^{i-1} \nu_h^{i-1} \right](s, \cdot) \right\|_\infty^2$$

$$\leq 2\|Q_h^i - Q_h^{i-1}\|_\infty^2 + 2H^2 \|\nu_h^i(\cdot \mid s) - \nu_h^{i-1}(\cdot \mid s)\|_1^2,$$

where the second line uses Holder's inequality and the fact that $\|Q_h^{i-1}\|_\infty \leq H$. In view of the update rule (3) for the $Q$-function, we further have

$$\|Q_h^i - Q_h^{i-1}\|_\infty = \left\| -\alpha_i Q_h^{i-1} + \alpha_i \left( r_h + \mathbb{P}_h \left[ (\mu_{h+1}^i)^\top Q_{h+1}^i \nu_{h+1}^i \right] \right) \right\|_\infty$$

$$\leq \alpha_i \max \left\{ \|Q_h^{i-1}\|_\infty, \left\| r_h + \mathbb{P}_h \left[ (\mu_{h+1}^i)^\top Q_{h+1}^i \nu_{h+1}^i \right] \right\|_\infty \right\}$$

$$\leq \alpha_i H.$$

As a result, we arrive at the bound

$$\mathrm{Err}_1 \leq 2\eta\alpha_t^1 \sum_{i=1}^{t} w_i \left(\alpha_i^2 H^2 + H^2 \|\nu_h^i(\cdot \mid s) - \nu_h^{i-1}(\cdot \mid s)\|_1^2\right)$$

$$= 2\eta H^2 \sum_{i=1}^{t} \alpha_t^i \alpha_i^2 + 2\eta H^2 \sum_{i=1}^{t} \alpha_t^i \|\nu_h^i(\cdot \mid s) - \nu_h^{i-1}(\cdot \mid s)\|_1^2,$$

where we again use the relation $w_i = \alpha_t^i/\alpha_t^1$. Since $\{\alpha_i\}_{i \leq t}$ is decreasing in $i$, we can apply Property 6 in Lemma 5 to obtain

$$\sum_{i=1}^{t} \alpha_t^i \alpha_i^2 \leq \frac{1}{t} \sum_{i=1}^{t} \alpha_i^2 \leq \frac{H+2}{t} \leq \frac{3H}{t},$$

where the second inequality follows from Property 5 in Lemma 5. In all, we see that

$$\mathrm{Err}_1 \leq \frac{6\eta H^3}{t} + 2\eta H^2 \sum_{i=1}^{t} \alpha_t^i \left\|\nu_h^i(\cdot \mid s) - \nu_h^{i-1}(\cdot \mid s)\right\|_1^2. \tag{14}$$

Substitute the upper bound (14) for $\mathrm{Err}_1$ into the master bound (12) to obtain

$$\mathrm{reg}_{h,1}^t(s) \leq \frac{\alpha_t \cdot (\log A)}{\eta} + \mathrm{Err}_1 - \mathrm{Err}_2$$

$$\leq \frac{2H \cdot (\log A)}{\eta t} + \frac{6\eta H^3}{t} + 2\eta H^2 \sum_{i=1}^{t} \alpha_t^i \|\nu_h^i(\cdot \mid s) - \nu_h^{i-1}(\cdot \mid s)\|_1^2$$

$$- \frac{1}{8\eta} \sum_{i=2}^{t} \alpha_t^{i-1} \|\mu_h^i(\cdot \mid s) - \mu_h^{i-1}(\cdot \mid s)\|_1^2,$$

where in the first inequality we use $\alpha_t = (H+1)/(H+t) \leq 2H/t$. Since $\|\nu_h^i(\cdot \mid s) - \nu_h^{i-1}(\cdot \mid s)\|_1 \leq 2$ and $\alpha_t^1 \leq 1/t$ (see Property 2 of Lemma 5), we can take the term $i = 1$ out and reach

$$\mathrm{reg}_{h,1}^t(s) \leq \frac{2H \cdot (\log A)}{\eta t} + \frac{16\eta H^3}{t} + 2\eta H^2 \sum_{i=2}^{t} \alpha_t^i \|\nu_h^i(\cdot \mid s) - \nu_h^{i-1}(\cdot \mid s)\|_1^2$$

$$- \frac{1}{8\eta} \sum_{i=2}^{t} \alpha_t^{i-1} \|\mu_h^i(\cdot \mid s) - \mu_h^{i-1}(\cdot \mid s)\|_1^2.$$

This finishes the proof of the regret bound (7a) for the max-player. The bound (7b) for the min-player can be obtained via symmetry.

Combine the two bounds (7a) and (7b) see that

$$\mathrm{reg}_{h,1}^t(s) + \mathrm{reg}_{h,2}^t(s) \leq \frac{2H \cdot \log(AB)}{\eta t} + \frac{32\eta H^3}{t}$$

$$+ \sum_{i=2}^{t} \left(2\eta H^2 \alpha_t^i - \frac{\alpha_t^{i-1}}{8\eta}\right) \left(\|\mu_h^i(\cdot \mid s) - \mu_h^{i-1}(\cdot \mid s)\|_1^2 \right. \tag{15}$$

$$\left. + \|\nu_h^i(\cdot \mid s) - \nu_h^{i-1}(\cdot \mid s)\|_1^2\right). \tag{16}$$

When $\eta \leq 1/(8H^2)$, one has

$$2\eta H^2 \alpha_t^i - \frac{\alpha_t^{i-1}}{8\eta} \leq 2\eta H^3 \alpha_t^i - \frac{\alpha_t^{i-1}}{8\eta} \leq -4\eta H^3 \alpha_t^i,$$

where we have used Property 3 of Lemma 5, i.e., $\alpha_t^{i-1}/\alpha_t^i \geq 1/H$. Consequently, with $\eta = C_\eta H^{-2}$ for some constant $C_\eta \leq 1/8$, the bound (16) reads

$$\max_s \left\{ \mathrm{reg}_{h,1}^t(s) + \mathrm{reg}_{h,2}^t(s) \right\} \leq \frac{3C_\eta^{-1} H^3 \cdot \log(AB)}{t} - 4\eta H^3 \sum_{i=2}^t \alpha_t^i \Big( \| \mu_h^i(\cdot \mid s) - \mu_h^{i-1}(\cdot \mid s) \|_1^2$$

$$+ \| \nu_h^i(\cdot \mid s) - \nu_h^{i-1}(\cdot \mid s) \|_1^2 \Big),$$

where we assume the choice of players is non-trivial, i.e., $AB \geq 2$.

### B.3   PROOF OF LEMMA 3

By Lemma C.2 in the paper by Zhang et al. (2022b), for any $h \in [H-1]$, we have the recursive relation

$$\delta_h^t \leq \sum_{i=1}^t \alpha_t^i \delta_{h+1}^i + \mathrm{reg}_{h+1}^t, \tag{17}$$

where we recall $\mathrm{reg}_{h+1}^t = \max_s \max_{i=1,2} \{ \mathrm{reg}_{h+1,i}^t(s) \}$.

**Step 1: Bounding $\mathrm{reg}_{h+1}^t$.**   In view of this recursion (17), one needs to control the maximal regret $\mathrm{reg}_{h+1}^t$ over the two players. Lemma 2 provides us with precise control of the individual regrets $\mathrm{reg}_{h,1}^t(s)$ and $\mathrm{reg}_{h,2}^t(s)$:

$$\mathrm{reg}_{h,1}^t(s) \leq \frac{3C_\eta^{-1} H^3 \cdot (\log AB)}{t} + 2\eta H^2 \sum_{i=2}^t \alpha_t^i \| \nu_h^i(\cdot \mid s) - \nu_h^{i-1}(\cdot \mid s) \|_1^2, \tag{18a}$$

$$\mathrm{reg}_{h,2}^t(s) \leq \frac{3C_\eta^{-1} H^3 \cdot (\log AB)}{t} + 2\eta H^2 \sum_{i=2}^t \alpha_t^i \| \mu_h^i(\cdot \mid s) - \mu_h^{i-1}(\cdot \mid s) \|_1^2, \tag{18b}$$

where we have substituted $\eta = C_\eta H^{-2}$ for $C_\eta \leq 1/8$ and $AB \geq 2$. We have also ignored the negative terms on the right hand sides of (7a) and (7b). Therefore, to control individual regrets, it suffices to bound the second-order path lengths $2\eta H^2 \sum_{i=2}^t \alpha_t^i \| \mu_h^i(\cdot \mid s) - \mu_h^{i-1}(\cdot \mid s) \|_1^2$ and $2\eta H^2 \sum_{i=2}^t \alpha_t^i \| \nu_h^i(\cdot \mid s) - \nu_h^{i-1}(\cdot \mid s) \|_1^2$. To this end, the following lemma proves crucial, whose proof is deferred to the end of this section.

**Lemma 6.** *For each $t, h$ and $s$, one has*

$$\mathrm{reg}_{h,1}^t(s) + \mathrm{reg}_{h,2}^t(s) \geq -2 \sum_{i=1}^t \alpha_t^i \delta_h^i.$$

In words, Lemma 6 reveals the approximate non-negativity of the sum of the regrets. This together with the upper bound (8) in Lemma 2 implies

$$2\eta H^2 \sum_{i=2}^t \Big( \alpha_t^i \| \mu_h^i(\cdot \mid s) - \mu_h^{i-1}(\cdot \mid s) \|_1^2 + \| \nu_h^i(\cdot \mid s) - \nu_h^{i-1}(\cdot \mid s) \|_1^2 \Big)$$

$$\leq \frac{3C_\eta^{-1} H^2 \cdot \log(AB)}{2t} + \frac{1}{H} \sum_{i=1}^t \alpha_t^i \delta_h^i.$$

Feeding this back to (18a) and (18b), we obtain

$$\text{reg}_h^t = \max_s \max_{i=1,2} \left\{ \text{reg}_{h,i}^t(s) \right\} \leq \frac{5C_\eta^{-1}H^3 \cdot \log(AB)}{t} + \frac{1}{H}\sum_{i=1}^t \alpha_t^i \delta_h^i. \tag{19}$$

**Step 2: Bounding $\delta_h^t$.**    Substituting the maximal regret bound (19) into the recursion (17), we arrive at

$$\delta_h^t \leq \left(1 + \frac{1}{H}\right)\sum_{i=1}^t \alpha_t^i \delta_{h+1}^i + \frac{5C_\eta^{-1}H^3 \cdot \log(AB)}{t}. \tag{20}$$

We continue the proof of Lemma 3 via induction on $h$. More precisely, we aim to inductively establish the claim

$$\delta_h^t \leq \sum_{h'=h}^H \left(1 + \frac{1}{H}\right)^{2(H-h')} \cdot \frac{5C_\eta^{-1}H^3 \cdot \log(AB)}{t}. \tag{21}$$

First note that the induction hypothesis holds naturally for $h = H$ as $\delta_H^t = 0$ for all $1 \leq t \leq T$. Now assume that the induction hypothesis is true for some $2 \leq h+1 \leq H$ and for all $1 \leq t \leq T$. Our goal is to show that (21) continues to hold for the previous step $h$ and for all $1 \leq t \leq T$. By the recursion (20) and the induction hypothesis, one has for any $1 \leq t \leq T$:

$$\delta_h^t \leq \left(1 + \frac{1}{H}\right)\sum_{i=1}^t \alpha_t^i \delta_{h+1}^i + \frac{5C_\eta^{-1}H^3 \cdot \log(AB)}{t}$$

$$\leq \left(1 + \frac{1}{H}\right)\sum_{i=1}^t \alpha_t^i \left(\sum_{h'=h+1}^H \left(1 + \frac{1}{H}\right)^{2(H-h')} \cdot \frac{5C_\eta^{-1}H^3 \cdot \log(AB)}{t}\right) + \frac{5C_\eta^{-1}H^3 \cdot \log(AB)}{t}.$$

Apply Lemma 4 to obtain

$$\sum_{i=1}^t \alpha_t^i \cdot \frac{5C_\eta^{-1}H^3 \cdot \log(AB)}{i} \leq \left(1 + \frac{1}{H}\right)\frac{5C_\eta^{-1}H^3 \cdot \log(AB)}{t}.$$

This leads to the conclusion that

$$\delta_h^t \leq \left(1 + \frac{1}{H}\right)\sum_{h'=h+1}^H \left(1 + \frac{1}{H}\right)^{2(H-h')}\left(1 + \frac{1}{H}\right)\frac{5C_\eta^{-1}H^3 \cdot \log(AB)}{t} + \frac{5C_\eta^{-1}H^3 \cdot \log(AB)}{t}$$

$$= \sum_{h'=h+1}^H \left(1 + \frac{1}{H}\right)^{2(H-h'+1)}\frac{5C_\eta^{-1}H^3 \cdot \log(AB)}{t} + \frac{5C_\eta^{-1}H^3 \cdot \log(AB)}{t}$$

$$= \sum_{h'=h}^H \left(1 + \frac{1}{H}\right)^{2(H-h')}\frac{5C_\eta^{-1}H^3 \cdot \log(AB)}{t}.$$

This finishes the induction.

This bound on $\delta_h^t$ can be further simplified by

$$\delta_h^t \leq \sum_{h'=h}^H \left(1 + \frac{1}{H}\right)^{2(H-h')} \cdot \frac{5C_\eta^{-1}H^3 \cdot \log(AB)}{t}$$

$$\leq H\left(1 + \frac{1}{H}\right)^{2H} \cdot \frac{5C_\eta^{-1}H^3 \cdot \log(AB)}{t}$$

$$\leq \frac{5e^2 C_\eta^{-1}H^4 \cdot \log(AB)}{t}.$$

This finishes the proof, and we are left with proving Lemma 6.

**Proof of Lemma 6.** Recall that

$$\text{reg}_{h,1}^t(s) + \text{reg}_{h,2}^t(s) = \max_{\mu^\dagger, \nu^\dagger} \sum_{i=1}^t \alpha_t^i \left[ \langle \mu^\dagger, Q_h^i \nu_h^i \rangle - \langle \nu^\dagger, (Q_h^i)^\top \mu_h^i \rangle \right](s).$$

Replace the estimation $Q_h^i$ with $Q_h^\star$ to obtain

$$\text{reg}_{h,1}^t(s) + \text{reg}_{h,2}^t(s) \geq \max_{\mu^\dagger, \nu^\dagger} \left[ \sum_{i=1}^t \alpha_t^i \left[ \langle \mu^\dagger, Q_h^\star \nu_h^i \rangle - \langle \nu^\dagger, (Q_h^\star)^\top \mu_h^i \rangle \right](s) \right.$$

$$\left. + \sum_{i=1}^t \alpha_t^i \left[ \langle \mu^\dagger, \left( Q_h^i - Q_h^\star \right) \nu_h^i \rangle - \langle \nu^\dagger, \left( Q_h^i - Q_h^\star \right)^\top \mu_h^i \rangle \right](s) \right].$$

Lower bounding the term involving $Q_h^i - Q_h^\star$ yields

$$\text{reg}_{h,1}^t(s) + \text{reg}_{h,2}^t(s) \geq \max_{\mu^\dagger, \nu^\dagger} \left[ \sum_{i=1}^t \alpha_t^i \left[ \langle \mu^\dagger, Q_h^\star \nu_h^i \rangle - \langle \nu^\dagger, (Q_h^\star)^\top \mu_h^i \rangle \right](s) \right] - 2 \sum_{i=1}^t \alpha_t^i \delta_h^i.$$

where recall $\delta_h^i = \|Q_h^i - Q_h^\star\|_\infty$. Now observe that $\sum_{i=1}^t \alpha_t^i \mu_h^i(\cdot \mid s)$ and $\sum_{i=1}^t \alpha_t^i \nu_h^i(\cdot \mid s)$ are valid policies, which implies

$$\max_{\mu^\dagger, \nu^\dagger} \left[ \sum_{i=1}^t \alpha_t^i \left[ \langle \mu^\dagger, Q_h^\star \nu_h^i \rangle - \langle \nu^\dagger, (Q_h^\star)^\top \mu_h^i \rangle \right](s) \right]$$

$$= \max_{\mu^\dagger, \nu^\dagger} \left[ \left\langle \mu^\dagger, Q_h^\star \left( \sum_{i=1}^t \alpha_t^i \nu_h^i \right) \right\rangle(s) - \left\langle \nu^\dagger, Q_h^{\star\top} \left( \sum_{i=1}^t \alpha_t^i \mu_h^i \right) \right\rangle(s) \right]$$

$$\geq \left\langle \left( \sum_{i=1}^t \alpha_t^i \mu_h^i \right), Q_h^\star \left( \sum_{i=1}^t \alpha_t^i \nu_h^i \right) \right\rangle(s) - \left\langle \left( \sum_{i=1}^t \alpha_t^i \nu_h^i \right), Q_h^{\star\top} \left( \sum_{i=1}^t \alpha_t^i \mu_h^i \right) \right\rangle(s)$$

$$= 0.$$

Combine the above two inequalities to finish the proof.

In the end, it is worth pointing out that without the improved inequality in Lemma 4, one would necessarily incur an extra $\log T$ factor in each induction step. Consequently, the recursion will fail due to the explosion at a rate of $(\log T)^H$.

### B.4 PROOF OF LEMMA 4

We prove the claim via induction. The base case $t = 1$ is true since $\alpha_1^1 \cdot 1 = 1 \leq 1 + 1/H$. Now assume that the inequality (9) holds for some $t \geq 1$, and we aim to prove that it continues to hold at $t + 1$. We first make the observation that for all $i \leq t$

$$\alpha_{t+1}^i = \alpha_i \prod_{j=i+1}^{t+1} (1 - \alpha_j) = (1 - \alpha_{t+1}) \alpha_i \prod_{j=i+1}^t (1 - \alpha_j) = (1 - \alpha_{t+1}) \alpha_t^i.$$

This allows us to rewrite $\sum_{i=1}^{t+1} \alpha_{t+1}^i \cdot \frac{1}{i}$ as

$$\sum_{i=1}^{t+1} \alpha_{t+1}^i \cdot \frac{1}{i} = (1 - \alpha_{t+1}) \left( \sum_{i=1}^{t} \alpha_t^i \cdot \frac{1}{i} \right) + \alpha_{t+1} \cdot \frac{1}{t+1}$$

$$\leq (1 - \alpha_{t+1}) \left( 1 + \frac{1}{H} \right) \frac{1}{t} + \frac{\alpha_{t+1}}{t+1},$$

where the second line follows from the induction hypothesis. Note that $\alpha_{t+1} = \frac{H+1}{H+t+1}$. We can continue the derivation as

$$\sum_{i=1}^{t+1} \alpha_{t+1}^i \cdot \frac{1}{i} \leq \left( 1 + \frac{1}{H} \right) \frac{t}{H+t+1} \cdot \frac{1}{t} + \frac{H+1}{H+t+1} \cdot \frac{1}{t+1}$$

$$= \left( 1 + \frac{1}{H} \right) \frac{t+1}{H+t+1} \cdot \frac{1}{t+1} + \left( 1 + \frac{1}{H} \right) \frac{H}{H+t+1} \cdot \frac{1}{t+1}$$

$$= \left( 1 + \frac{1}{H} \right) \frac{1}{t+1}.$$

This finishes the proof.

$\square$