# OpenReview forum: "$O(T^{-1})$ Convergence of Optimistic-Follow-the-Regularized-Leader in Two-Player Zero-Sum Markov Games "
_ICLR.cc/2023/Conference — ICLR 2023 poster_

### Official Review · Reviewer_j4CZ · 2022-10-17

**Confidence:** 4
**Correctness:** 4
**Technical Novelty And Significance:** 3
**Empirical Novelty And Significance:** Not applicable
**Recommendation:** 6

**Clarity, Quality, Novelty And Reproducibility:**

I think overall the work is clearly presented. The main technique leading to the improved convergence rate is new. The proof seems correct to me upon a quick skimming.

**Strength And Weaknesses:**

Strengths:

* The paper settles the open problem raised by Zhang et al. (2022) about $T^{-1}$ convergence rate of OFTRL in Markov Games. In particular, the same work designed a modified OFTRL algorithm achieving $\tilde{O}(T^{-1})$ rate, and also showed that empirically the original OFTRL algorithm does achieve $T^{-1}$ rate. Plus, $T^{-1}$ convergence in the easier case of zero-sum normal-form games (NFGs) is a well-established result (e.g. Rakhlin and Sridharan, 2013) and follow directly from the RVU property of OFTRL. These all seem to be urging for a positive resolution to this problem, which this paper did.

* The main technique—establishing an *approximate* non-negativity of the summed regret for zero-sum NFGs with slowly changing game matrices—seems interesting and new to this line of work. As the authors mentioned, the technique of using non-negativity of regrets to establish second-order path length bounds has been used in other contexts such as fast convergence of swap regrets. However, to my best knowledge, this is its first application in Markov Games.

* The topic of fast convergence in games should be of good interest to the RL/games community.

Weaknesses:

* Overall, the paper sounds a bit lacking in terms of completeness as for a conference paper on this topic (only one main theorem and no extensions / additional results etc). I wonder if the authors have thought about whether the techniques (approximate non-negative regret) or settings (zero-sum NFGs with slowly changing game matrices) could be useful in other problems? (For example, any relation to “time-changing zero-sum games”? see ref below). Having some additional results about related problems or techniques could genuinely improve the paper, in my opinion.
Zhang, M., Zhao, P., Luo, H., & Zhou, Z. H. (2022). No-Regret Learning in Time-Varying Zero-Sum Games. arXiv preprint arXiv:2201.12736.

* The claim of settling Zhang et al. (2022)’s open question needs to be discussed carefully, as that work also gives a modified OFTRL algorithm with the same $\tilde{O}(T^{-1})$ rate. Currently this is discussed only in Section 3, and I suggest also adding some discussions in the introduction/related work where appropriate.


**Summary Of The Paper:**

This paper studies the convergence of optimistic policy optimization algorithms in two-player zero-sum Markov Games. The main result of the paper is to show that the Optimistic-Follow-The-Regularized-Leader (OFTRL) algorithm achieves an $O(T^{-1})$ convergence rate to the Nash Equilibria of the game. This settles the open problem raised in the recent work of (Zhang et al. 2022), which showed an $\tilde{O}(T^{-5/6})$ convergence rate for the same algorithm.

**Summary Of The Review:**

The paper resolves an open question on the fast convergence in Markov Games using an interesting new technique, though the result sounds a bit thin and lacking in extensions / additional results.

---

> ### Author Response · Authors · 2022-11-14
> **Authors' response to the review of Reviewer j4CZ**
>
> We thank the reviewer for the nice summary of our paper and our contributions. We also thank the reviewer for bringing the time-varying setting to our attention. Our technique of approximate non-negativity is somewhat similar to what they have done in the proof of Theorem 4 (lower bounding the sum of regret by $W_T:=\sum_{t=1}^T ||A_t-\bar{A}||_\infty$). However, the difference lies in that the non-negativity gap in Markov games comes from the estimation error of the equilibrium pay-off matrix ($||Q_h^t - Q_h^\star||_\infty$). This estimation error also decays at the rate of $O(T^{-1})$ as it can be further controlled by the regret in future horizons, while $W_T$ depends solely on the payoff matrix sequence $\{A_t\}$. We will add a discussion on this in the paper.
>
> Regarding the comparison with the modified OFTRL algorithm, we will add a more careful discussion on it in the revision.

---

### Official Review · Reviewer_krEs · 2022-10-24

**Confidence:** 4
**Correctness:** 4
**Technical Novelty And Significance:** 1
**Empirical Novelty And Significance:** Not applicable
**Recommendation:** 6

**Clarity, Quality, Novelty And Reproducibility:**

The proposed algorithms and proof are not novel given the existing work Zhang et al., 2022.

**Strength And Weaknesses:**

Pros:
- The presentation is clear.
- The proof is correct.

Cons:
- It lacks novelty for either the algorithm design or the proof. The proposed algorithm itself actually can be regarded as a special realization of  Algorithm 1 in Zhang et al. 2022, which makes the algorithm design less important. The proof also reuses lots of statements in Zhang et al., 2022 (like the proof of Lemma 1). Finally, the proposed algorithm enjoys the nearly same order of convergence rate as Algorithm 10 in Zhang et al., and I am not convinced that the proposed algorithm is simpler than Algorithm 10 in Zhang et al. since it is already simple enough.

**Summary Of The Paper:**

This paper studies solving two-player zero-sum Markov games using follow-the-regularized-leader-type algorithms. The authors propose an algorithm based on Zhang et al., 2022 and proved it has an $O(1/T)$ convergence rate to the Nash equilibrium.

**Summary Of The Review:**

The authors propose an algorithm for two-player zero-sum Markov Games with an $O(1/T)$ convergence rate to find the Nash equilibrium. I find the proposed algorithm and theory lack novelty and importance given Zhang et al., 2022, therefore I recommend a reject.

---

> ### Author Response · Authors · 2022-11-14
> **Authors' response to the review of Reviewer krEs**
>
> We thank the reviewer for reading our paper and providing feedback. However, we disagree with the comment: ''It lacks novelty for either the algorithm design or the proof''.
>
>
> First, we'd like to clarify that we do not claim the novelty of the algorithm at all. The OFTRL algorithm with smooth $Q$-value updates is proposed in Zhang et al. [1], and we have properly acknowledged this fact in our paper.
>
> Second, we'd like to emphasize that the main contribution of this paper is on the proof part: we settle the open problem raised by Zhang et al. [1] about the convergence rate of OFTRL in Markov Games. In this reference, Zhang et al. showed empirically that the OFTRL algorithm achieves an $O(T^{-1})$ rate, while they only manage to establish a worse $\widetilde{O}(T^{-5/6})$ rate. We close this gap by proving the improved $O(T^{-1})$ rate. To obtain the improved rate, we have two technical innovations. First, we establish an approximate non-negativity of the summed regret for zero-sum Markov games. To the best of our knowledge, this is new to this line of research in Markov games. This allows us to obtain an improved stability bound on the iterates of the algorithm, compared to that provided in Zhang et al. [1]. Second, we refine the analysis of the weights used in OFTRL that shaves an extra $\log(T)$ factor (Lemma 4). This is a crucial improvement that allows a *different* analysis of the recursion in the proof of Lemma 3 from the one used in Zhang et al. [1]. It avoids an exponential explosion along the $H$ horizons, which would create an undesired factor of $(\log T)^H$ if we follow the strategy in Zhang et al. [1]. These two technical novelties are both critical in settling the open problem.
> Indeed in the proof, we reuse some of the statements in Zhang et al. [1] to simplify the writing. However, this does not diminish the importance of the technical contributions we make in the current paper.
>
> Lastly, we'd like to comment on the *modified* OFTRL algorithm that was proposed by Zhang et al. [1] as a remedy to the original OFTRL algorithm to achieve the $\widetilde{O}(T^{-1})$ rate. Given that the *modified* OFTRL algorithm achieves the desired $\widetilde{O}(T^{-1})$ rate, we still think it is important to establish the fast convergence of the original OFTRL for the following reasons. First, the original OFTRL falls into the general algorithm design framework for solving Markov games proposed Zhang et al. [1], while the modified OFTRL algorithm does not. This fact was also acknowledged by Zhang et al. [1]. Second, in Zhang et al. [1], there exists a gap between the theoretical prediction $\widetilde{O}(T^{-5/6})$ of OFTRL, and its empirical performance ${O}(T^{-1})$, which urges for a positive resolution. On a minor note, the update to $Q$-function in the modified OFTRL algorithm can be more complicated, as the modified OFTRL takes maximum in all policies versus a fixed policy in the original OFTRL. In tabular games this would be a minor issue, but it may be costly in certain query models of the $Q$-function.
>
> Reference
>
> [1] Runyu Zhang, Qinghua Liu, Huan Wang, Caiming Xiong, Na Li, and Yu Bai. Policy optimization for
> markov games: Unified framework and faster convergence. arXiv preprint arXiv:2206.02640, 2022.

---

> > ### Comment · Reviewer_krEs · 2022-12-13
> > **Thanks for the response**
> >
> > I have read the authors' responses and other reviewers' comments. I am now convinced that the authors made non-trivial contributions to Zhang et al. according to their new regret analysis. Therefore, I tend to raise my score to 6.

---

### Official Review · Reviewer_ceBg · 2022-10-31

**Confidence:** 4
**Correctness:** 4
**Technical Novelty And Significance:** 3
**Empirical Novelty And Significance:** Not applicable
**Recommendation:** 6

**Clarity, Quality, Novelty And Reproducibility:**

I think the paper is well written, has merits technically and the result is important.

**Strength And Weaknesses:**

The result is important and adds value to the current growing literature. The techniques are interesting, exploiting ideas from papers by Anagnostides et al that the sum of regrets should be non-negative (in this case are not too negative) and moreover this leads to prove that second-order path length is bounded. This idea can give a log T/T bound but the authors with extra effort remove the log T dependence.

There is another work (appeared a bit later I think) that achieves 1/T convergence using Optimistic MWU and the notion QRE (quantal response) Nash. The ideas are somewhat different as in the latter the KL divergence is used as a potential.

**Summary Of The Paper:**

The paper focuses on two player zero-sum stochastic games. It is shown that Optimistic FTRL (together with value update step) converges to a Nash equilibrium in $O(1/T)$ improving the result of Zhang et al.

**Summary Of The Review:**

I think the paper is above the bar for acceptance. If there was a score 7, I would put 7, between 6 and 8 I choose 6 but I am in favor of the paper getting in. The only reason I chose 6 is that there have appeared many papers of the same flavor/similar results the last year and reading this paper did not make me feel surprised. If needed, I will increase my score.

Correction: After checking thoroughly the paper by Zhang et al, there is an improved analysis that also achieves O(log T/T) regret, so the improvement is not as large as was mentioned in the paper. Can you argue about that? As far as I can see now, the result improves from \log  T/T to 1/T regret. The comment/question has been addressed.

---

> ### Author Response · Authors · 2022-11-14
> **Authors' response to the review of Reviewer ceBg**
>
> We thank the reviewer for the nice words about our paper. As a response to the reviewer's correction bullet, we'd like to clarify that the improved rate $\widetilde{O}(T^{-1})$ in Zhang et al. [1] was not established for the original OFTRL algorithm we consider herein, but for a *modified* OFTRL algorithm. Zhang et al. [1] only proved an $\widetilde{O}(T^{-5/6})$ for the original OFTRL algorithm, which we improve to $O(T^{-1})$ in this paper. This settles the open problem raised in Zhang et al. [1] on establishing such a fast convergence rate for original OFTRL. We would also like to point out that the *modified* algorithm diverges from the general framework proposed in Zhang et al. [1]. Therefore it is important to show an $O(T^{-1})$ rate of the unmodified one. This also matches the $O(T^{-1})$ empirical rate observed in Zhang et al [1]. Let us know if this addresses your concern.
>
>
> Reference
>
> [1] Runyu Zhang, Qinghua Liu, Huan Wang, Caiming Xiong, Na Li, and Yu Bai. Policy optimization for
> markov games: Unified framework and faster convergence. arXiv preprint arXiv:2206.02640, 2022.

---

> > ### Comment · Reviewer_ceBg · 2022-11-17
> > **Thanks for the response**
> >
> > Dear authors,
> >
> > Thank you for your response. Please add this explanation in the introduction when you talk about two player zero sum stochastic games (i.e., that a modified optimistic ftrl with entropy regularizer gives log T/T proposed by Zhang et al. Please also add the fact that Zhao et al paper gives last iterate convergence as well.

---

> > > ### Author Response · Authors · 2022-11-19
> > > **Update on the submission**
> > >
> > > Thank you for your suggestions. We have posted an update to our submission. It addresses the things you mention in the introduction.

---

### Decision · Program_Chairs · 2023-01-20

**Decision:**

Accept: poster

**Justification For Why Not Higher Score:**

The results received only mild support from the reviewers. The setting is the easiest variant of zero-sum Markov games, and the novelty and potential impact of the paper is somewhat limited.

**Justification For Why Not Lower Score:**

All reviewers support (mildly) the acceptance of the paper, which provides an improvement compared to earlier analysis.

**Metareview: Summary, Strengths And Weaknesses:**

The paper presents an analysis showing that optimistic FTRL combined with online averaging in the value-function updates finds an $O(T^{-1})$ Nash equilibrium in $T$ iterations for two-player zero-sum Markov games with known rewards and transition dynamics. This improves upon previous analysis of the same algorithm. Also note that the resulting bound scales worse in the horizon $H$ than the tweaked version presented in earlier work which also achieves $\tilde O(T^{-1})$ convergence.

**Note From Pc:**

if the above contains the word "oral" or "spotlight" please see: "oral" presentation means -> notable-top-5% and "spotlight" means -> notable-top-25%. As stated in our emails, we are disassociating presentation type from AC recommendations